# Linking Grape Cell Wall Composition and Phenolic Release to Wine Quality: Effects of Methyl Jasmonate-Loaded Chitosan Nanoparticles in Monastrell

**DOI:** 10.3390/plants14243817

**Published:** 2025-12-15

**Authors:** Rocío Gil-Muñoz, Juan Daniel Moreno-Olivares, María José Giménez-Bañón, José Julián Pérez-Cuadrado, María Quílez-Simón, Eva Pilar Pérez-Álvarez, Teresa Garde-Cerdán, Luis Javier Pérez-Prieto, Antonio Abel Lozano-Pérez

**Affiliations:** 1Equipo de Enología y Viticultura, Instituto Murciano de Investigación y Desarrollo Agrario y Medioambiental, Jumilla 30520, Spain; juand.moreno5@carm.es (J.D.M.-O.); mariaj.gimenez8@carm.es (M.J.G.-B.); luisjavier.perez@carm.es (L.J.P.-P.); 2Grupo de Nanotecnología, Instituto Murciano de Investigación y Desarrollo Agrario y Medioambiental, Murcia 30150, Spain; josej.perez3@carm.es (J.J.P.-C.); maria.quilez@carm.es (M.Q.-S.); antonioa.lozano@carm.es (A.A.L.-P.); 3Instituto de Ciencias de la Vid y del Vino (Gobierno de La Rioja-CSIC-Universidad de La Rioja), Carretera de Burgos Km. 6, Finca La Grajera, 26007 Logroño, Spain; evapilar.perez@icvv.es (E.P.P.-Á.); teresa.gardecerdan@gmail.com (T.G.-C.)

**Keywords:** wine quality, methyl jasmonate, grapevine, phenolic compounds, cell wall composition, chitosan nanoparticles

## Abstract

This study explored the effects of three elicitor treatments—methyl jasmonate (MeJ), chitosan nanoparticles (ChNPs), and their combination (MeJ-ChNPs)—on cell wall characteristics in Monastrell grapes during the 2024 vintage. The central aim was to determine whether incorporating MeJ into chitosan nanoparticles could reduce its volatility, enhance its stability, and ultimately strengthen its elicitation effect when applied in vineyards. By applying these formulations, particularly the MeJ-ChNPs complex, the research sought to understand how grape maturation, cell wall morphology, and phenolic composition could be altered, and how these changes might translate into differences in wine quality. The implementation of these MeJ formulations, and particularly MeJ-ChNPs, resulted in delayed grape maturation and pronounced changes in cell wall morphology, including increased thickness and altered phenolic composition. Grapes treated with MeJ-ChNPs showed enhanced anthocyanin biosynthesis and distinct colour properties compared to untreated controls. The findings revealed that MeJ-ChNPs delayed grape ripening and induced notable modifications in cell wall structure, including increased thickness and shifts in phenolic composition. These structural changes influenced the extractability of phenolic compounds and shaped the chromatic attributes of the resulting wines. Multivariate analyses, including principal component and correlation analyses, highlighted clear differences amongst treatment groups, emphasising the effectiveness of nanoparticle-based elicitors in changing grape skin morphology and composition and improving wine quality.

## 1. Introduction

The foliar application of different compounds that act as elicitors is a strategy that can be used to improve grape quality. Amongst chemical elicitors, MeJ is a natural phytohormone that has been reported to upregulate endogenous levels of health-promoting compounds by increasing phenolics, anthocyanins, and flavonoid biosynthesis, in addition to increasing antioxidant levels [1]. This elicitor can activate the enzymes that are responsible for the biosynthesis of polyphenols, such as phenylalanine ammonia-lyase (PAL) [2]. Additionally, MeJ has been shown to exhibit a synergistic effect in maintaining higher firmness and delaying the activities of cell wall hydrolysing enzymes in blueberries (*Vaccinium ashei*) [3], mandarins (*Citrus reticulata* × *Citrus sinensis* ‘Kinnow’) [4] and loquats (*Eriobotrya japonica* Lindl.) [5]. However, due to the volatility and low solubility of this phytohormone, high MeJ concentrations are required, which increases costs and is not profitable for the agricultural sector.

On the other hand, chitosan (Ch) is a linear co-polymer composed of β-(1–4)-linked N-acetyl-D-glucosamine obtained by partial deacetylation of chitin [6]. In recent years, chitosan, which has also been used as a possible elicitor due to its unique combination of properties, such as biocompatibility, biodegradability, metal complexation, and antimicrobial activities [7], has gained attention. Therefore, chitosan derivatives can be ideal as a natural polymer for the induction of plant defence mechanisms (i.e., different defence metabolic pathways) that may eventually contribute to increased plant growth, protection, and crop productivity [8].

Given the recent and increasing trend towards sustainable and efficient agriculture, the application of nanotechnology in agriculture is expanding, with notable developments in viticulture. Nanoparticles are materials of small size (1–100 nm) that have a large surface area, can be used for the targeted and controlled release of nutrients, with greater efficiency, and minimised losses, increasing their availability to plants [9]. According to our experience in using MeJ as a grape quality improver due to the increase in secondary metabolites and the properties of chitosan, chitosan nanoparticles (ChNPs) loaded with MeJ could address challenges such as volatility, low solubility, and instability, thereby improving the delivery of plant growth regulators. The small size and large surface area of ChNPs could make them more effective than bulk chitosan in enhancing plant growth and defence under abiotic stress. Recent studies have demonstrated that some types of ChNPs not only provide protection against phytopathogens and promote growth, but also play an essential role in physiological traits, protection, secondary metabolites, and growth [7].

On the other hand, the cell wall undergoes remodelling throughout various developmental stages and in response to external stimuli. Research on the cell wall integrity maintenance system suggests that plants respond to different stress conditions with adaptive mechanisms, including increased cell wall thickness [10]. These findings imply that the treatment itself might have induced stress in the plant, triggering the thickening of the fruit’s cell walls. Indeed, multiple studies have shown that various forms of stress could compromise cell wall integrity. This phenomenon is associated with the generation of reactive oxygen species (ROS) and the enhanced activity of peroxidases, xyloglucan-modifying enzymes, and expansins [11]. Furthermore, high nitrogen levels have been shown to suppress the expression of genes involved in lignin and cellulose biosynthesis in rice roots [12].

Cell walls are integral components of grape skins, and although the skin accounts for only a small fraction of the total berry weight, it plays a crucial role in determining wine quality, as it contains the majority of aromatic and phenolic compounds. Consequently, understanding the composition and structural characteristics of grape skin is essential, as these factors influence the berries’ mechanical resistance, texture, and processing efficiency [2]. The cell walls of grape skin are particularly significant due to their highly complex and dynamic nature. They are composed of polysaccharides, phenolic compounds, and proteins, stabilised by both ionic and covalent bonds. These components can vary not only between grape varieties [13] but also within the same variety grown in different terroirs [14]. Such structural diversity has important technological implications, as it affects the extractability of key compounds during winemaking, ultimately shaping the final quality of the wine.

Taking the above into account, the aim of this research was to evaluate the effects of MeJ treatments applied conventionally and in chitosan nanoparticles loaded with MeJ on the structural components of the cell walls of Monastrell grape skins, together with their impact on the extraction of phenolic compounds in the resulting wines.

## 2. Results and Discussion

### 2.1. Preparation and Characterisation of Methyl Jasmonate-Loaded Chitosan Nanoparticles

To have sufficient volume of nanoparticle formulation for field treatments, 4.8 L of each stock aqueous suspension (5×) was prepared and characterised, ready to be diluted with water. On one hand, the suspension of MeJ-ChNPs at 5 mg/mL of chitosan and 10 mM of MeJ was successfully prepared for the experimental treatment following the procedure described in Figure 1A. On the other hand, an equivalent batch of unloaded ChNPs was prepared without MeJ as a negative control.

The determination of the hydrodynamic characteristics by dynamic light scattering (DLS) revealed that both suspensions of nanoparticles, either the MeJ-ChNPs or the unloaded ChNPs prepared as a control, are stable due to their small size and high value of zeta potential (ζ), and slightly varied after the incorporation of MeJ into the formulation. The addition of MeJ produced a reduction in the average diameter (Zave) and the polydispersity index (PdI) of the nanoparticles from 306.5 ± 17.4 nm and 0.388 ± 0.062 for the ChNPs to 236.3 ± 17.0 nm and 0.255 ± 0.070 for the MeJ-ChNPs, as can be seen in Figure 1B. Nevertheless, the ζ was not significantly affected by the MeJ loading, showing values ranging from 39.9 ± 1.8 mV for ChNPs to 38.7 ± 1.7 mV for MeJ-ChNPs, as can be seen in Figure 1C. These high values of ζ provide a high stability of the nanoparticle suspensions due to the electrostatic repulsions of positive charges. It is noteworthy that a similar range of ζ has been found in other studies, where ChNPs were formulated and loaded with different plant extracts [15] or essential oils [16]. As can be seen in Figure 1D,E, scanning electron micrographs of the nanoparticles at different magnifications showed that either the ChNPs or the MeJ-ChNPs are spherical in shape with sizes lower than those observed by DLS due to the dehydration/metallisation process carried out for sample preparation.

### 2.2. Quality Parameters in Grape Fruit

The physico-chemical data for grapes from the control and the different applied treatments at the time of harvest are shown in Table 1. Some differences were found in °Brix at harvest. The highest value was shown in control grapes, which were the most mature berries. However, the treated grapes exhibited lower values of sugar content at harvest, indicating a delayed ripening effect in the treated grapes. These results are in agreement with those found by other authors, such as Giménez-Bañón et al. [17], who showed a delay in the maturation of Monastrell grapes treated with urea nanoparticles, or Paladines-Quezada et al. [2], who, in a study carried out on several grape varieties, observed how BTH (benzothiadiazole) and MeJ treatments decreased the °Brix in Monastrell in one of the studied vintages. Wang et al. [3] also found a reduction in °Brix, glucose, and fructose after MeJ treatment in the Gewürztraminer grape variety, pointing to an elicitor-repressive effect on berry maturation. This fact may be essential to reduce the imbalance between phenolic and technological maturity that this variety suffers because of climate change in arid areas, such as southeastern Spain. However, other authors, including Garde-Cerdán et al. [18], showed no differences between control grapes and grapes treated with MeJ in conventional or nanoparticle form applied to the Tempranillo variety.

Regarding total acidity, all of the treatments showed higher values in these parameters compared to the control. This is aligned with the results obtained by other authors, such as Gil-Muñoz et al. [19], who in a maturation study showed that control grapes demonstrated a more accelerated decrease in acidity than grapes treated with MeJ in conventional or nanoparticle form. Other studies have also reported an increase in total acidity at harvest time in Monastrell grapes [20,21] or Tempranillo grapes when MeJ was applied [22]. However, D’Onofrio et al. [23] observed that MeJ treatment diminished the total acidity of the Sangiovese grape variety. In contrast, Giménez-Bañón et al. [17] found no statistical differences for this parameter when Monastrell grapes were treated with urea or nano-urea in any of the studied campaigns.

Concerning pH, due to the climatology that can be found in this part of Spain, the pH of the musts is usually close to 4; nevertheless, in this study, we were able to observe a decrease in this value for all of the treated samples, with the grapes treated with MeJ-ChNPs obtaining the lowest value. This fact is of great importance, as high-pH grape juice may generate technical problems with complex solutions during alcoholic fermentation [24]. Tartaric acid and malic acid are the main acids in grapes, and can be found among the acids with a higher contribution to the acidity parameter of musts. Our results showed an increase in tartaric acid in all of the treatments compared to control grapes; however, malic acid only showed higher values in grapes treated with MeJ.

Finally, berry size is an essential parameter in winemaking, as it provides information on the skin area/berry volume ratio. Control grapes reached the highest berry weight, followed by MeJ- and MeJ-ChNPs-treated grapes, which reached intermediate values, and, finally, ChNPs grapes exhibited the lowest berry weight. On the contrary, in a study carried out in Monastrell treated with urea in conventional form and in nanoparticles, Giménez-Bañón et al. [17] showed that the treatments did not influence this parameter. The same was demonstrated by authors such as Hannam et al. [25] or Lasa et al. [26], who found that berry size was not influenced by any of the parameters and did not detect a significant increase in yield in grapes foliar-treated with urea.

### 2.3. Extractability Parameters

With regard to the assessment of phenolic compound extractability, it is strongly influenced by the extraction method that is used. In this sense, the cellular maturity index (CMI), defined by Glories and Augustin [27], appears to provide adequate robustness in predicting phenolic compounds in the resulting wines [28].

The results for the extractability parameters are shown in Table 2. As can be seen, the highest values of extractable anthocyanins and polyphenols (corresponding to a pH of 3.6) were found in the control grapes. However, when we take into account total anthocyanins (extracted at a pH of 1), the highest values were observed in grapes treated with MeJ-ChNPs. These results indicate that, in the case of treated grapes, it would be necessary to apply some technology during vinification to break the cell wall integrity and release all the anthocyanin content, since the treatments increased the phenolic composition. One proposal could be the use of enzymes during winemaking, as suggested by Castro-Lopez et al. [29], who stated that maceration enzymes might be used for improving must volume and clarification, enhancing filterability, and, especially in red winemaking processes, increasing the degradation of the skin cell walls, the limiting barrier for the extraction of phenolic compounds.

Concerning the CMI, the highest values were observed in the grapes treated with nanoparticles, indicating a greater difficulty in the extraction of anthocyanins during the winemaking process. Extractability is considered good when the difference between ApH3.6 and ApH1 is slight, and therefore, the extractability index is low [13]. Two facts could be responsible for a higher CMI in nanoparticle-treated grapes: a lower maturity, something that we could observe in all of the treatments at harvest time (Table 1) compared to the control grapes; or the fact that the elicitation effect can produce a greater thickening of the cell wall, making it challenging to extract anthocyanins during the winemaking process. Authors such as Hernández-Hierro et al. [30] demonstrated that differences in anthocyanin extractability were highly influenced by the ripeness degree and, to a lesser extent, by the soluble solids content. Conversely, other studies on cell wall integrity (CWI) maintenance mechanisms suggest that exposure to diverse stress conditions elicits a range of adaptive responses, including the reinforcement of the cell wall through increased thickness [10].

Regarding the SMI, no significant differences were found between the control and treated grapes, which indicates that the contribution of tannins by the seed remained comparable across treatments. Authors such as Ribéreau-Gayon et al. [31] reported that, in skins, phenolic extractability increases with maturity, whereas a decrease is observed in seeds.

Therefore, the phenol extractability indices are key factors in wine grape quality, influencing the winemaking methodology chosen to ensure the maximum release of phenolic compounds.

### 2.4. Cell Wall Composition

Table 3 shows the amount of skin and cell wall material (CWM) isolated from the different control and treated grapes. The highest percentage of skin was observed in the control grapes and the grapes treated with ChNPs. In contrast, the lowest percentage of skin corresponded to MeJ- and MeJ-ChNPs-treated grapes, which could be a disadvantage from an oenological point of view, since the grape skin is the primary source of colour and aroma compounds.

In relation to the isolated CWM, these values were higher than those shown by other authors who also studied Monastrell grapes in the same [14,32]. In terms of the treatments, control grapes showed the lowest amount of CWM in the skins compared to treated grapes, with MeJ-ChNPs following as the ChNPs-treated grapes that presented the largest quantities. The most considerable quantities of CWM found in treated grape skins could indicate a more substantial barrier for the extraction of the compounds located inside these cells, as previously stated by Ortega-Regules et al. [13].

If we take into account the relationship between the amount of skin and the calculated cell wall, we can see that the lowest values are obtained in the treatments with nanoparticles, indicating that there is a greater amount of cell wall in these two treatments compared to the control grapes and even with the treatment with MeJ. This fact could be attributed to a greater thickening of the cell wall generated by these nanoparticles, as already mentioned in previous sections. In a study carried out for several years applying urea to Monastrell in conventional and nanoparticle forms, Giménez-Bañón et al. [17] observed an increase in CWM in each of the treatments compared to that obtained in the control grapes, indicating a thickening of the cell walls in the urea and nano-urea treatments. These facts could indicate that the application of the treatments produced some stress to the plant, which induced the thickening of the cell walls of the berries. On the contrary, Paladines-Quezada et al. [2,32], in two separate studies conducted in 2015 and 2017 on Monastrell grapes, found no differences regarding the amount of skin in control and MeJ-treated grapes.

### 2.5. Carbohydrate Composition in Cell Walls

The carbohydrates found in the skin cell wall indicated the presence of pectic polysaccharides (as uronic acids), hemicellulose and cellulose. Cellulosic glucose and uronic acids accounted for the highest percentages of sugars in the cell walls of the skin, as described in the same varieties by Ortega-Regules et al. [33]. In our study, the three parameters related to carbohydrate (uronic acids, hemicellulose, and cellulose) content showed the highest values in the composition of the analysed cell wall. Previous studies have shown that the cell wall material from grape mesocarp and exocarp consists mainly of cellulose and pectic polysaccharides [34]. As can be seen, Table 4 shows the results corresponding to the grape skin cell composition in control and treated grapes. The carbohydrate composition of the skin cell walls of the different grapes depended on the treatment.

Uronic acids (UAs) are recognised as heteropolysaccharides primarily composed of galacturonic acid residues, which may occur as methoxyl esters and include varying amounts of neutral sugars present as side chains [35]. This pectic fraction consists mainly of 65% homogalacturonan, 23% arabinogalactan types I and II, 10% rhamnogalacturonan type I, and 2% rhamnogalacturonan type II [34,36]. Regarding the results obtained for this parameter (Table 4), the highest concentrations were obtained in grapes treated with MeJ-ChNPs. This is in accordance with the results obtained by Giménez-Bañón et al. [17] who observed an increase in UA content in Monastrell with both treatments of 69% (urea) and 76% (nano-urea) in 2020, and 47% (urea) and 27% (nano-urea) in 2021. In a study carried out in the same variety, the same authors observed an increase in UA when grapes were treated with MeJ or MeJ nanoparticles that was over 40% in 2020 and 2021; nevertheless, no increase was recorded in 2019. Finally, Paladines-Quezada et al. [2] also observed an increase in UAs in Merlot grapes treated with MeJ and BTH.

Cellulose and hemicellulose are classified as neutral polysaccharides. Cellulose microfibrils are composed of glucose units linked by β-(1→4) glycosidic bonds, forming cellobiose chains [37]. Hemicellulose interacts with cellulose microfibrils, contributing to the structural integrity of the cell wall. In type I cell walls, hemicellulose is predominantly present as xyloglucan, with smaller proportions of galactoglucomannans, galactomannans, glucans, and glucuronoarabinoxylans [38]. In our study, hemicellulose levels were highest in control grapes compared to those subjected to treatment (Table 4). The presence of hemicellulose is indicative of the occurrence of hemicellulosic polysaccharides [13], and an increase in the amount of hemicellulose would lead to a strengthening of the cell wall, as it acts by fixing the cellulose microfibrils [33]. Authors such as Castro-López et al. [29] stated that ripe skin cell walls showed the highest values of hemicellulose, which is in agreement with our results, as the control grapes turned out to be the ripest at the time of harvest. However, Paladines-Quezada et al. [32] observed that MeJ and MeJ + BTH treatments caused a decrease in the hemicellulose concentration of Monastrell grapes during two consecutive seasons.

In relation to cellulose results (Table 4), higher values could also be observed in the grapes treated with MeJ, both in conventional form and in nanoparticle form, compared to control grapes. ChNPs-treated grapes showed intermediate values between control grapes and those treated with MeJ (conventional or nanoparticles). Different authors have shown similar findings for this parameter in heterogeneous treatments applied to Monastrell grapes. Giménez-Bañón et al. [17] observed that control grapes generally exhibited the highest values compared to urea- and nano-urea-treated grapes. These results would be relevant from a technological point of view, since an increase in the amount of cellulose, together with a thicker skin, may explain the difficulties that can usually be observed in the extraction of anthocyanins from Monastrell grapes during winemaking [28], given that cellulosic glucose quantities have been correlated with firmness.

### 2.6. Total Phenols and Proteins in Cell Walls

In grape skin, polyphenol compounds are bound to polysaccharides by hydrophobic interactions and hydrogen bonds, and are components of both the primary and secondary cell walls [39]. These compounds establish phenolic cross-links with other cell wall constituents such as extensins, lignins, glucuronoarabinoxylans, and rhamnogalacturonan-I side chains. Cross-linking modifies wall consistency, cell expansion, and pathogen resistance [40].

Regarding the phenolic compound results (Table 4), the highest values were found in the control grapes. The composition of total phenols was significantly higher in control grapes compared to the treated grapes, yielding values that were twice as high as those found in the treated grapes. A large part of the detected polyphenolic compounds could be tannins, as tannins integrated in the skin cell walls have been previously detected. Amrani Joutei et al. [41] and Nunan et al. [36] also revealed a high content of phenolics in skin cell walls. As can be seen in Section 2 of this paper, related to extractability parameters, the highest concentration of total polyphenols is observed at a pH of 3.6 in the control grapes. Different findings have been shown for this parameter. Paladines-Quezada et al. [2] reported that methyl jasmonate (MeJ) increased the concentration of phenolic compounds in the cell walls of Monastrell and Cabernet Sauvignon grape varieties during the 2015 and 2016 vintages. However, in the Merlot variety, both MeJ and benzothiadiazole (BTH) treatments led to a reduction in these compounds across both years. Conversely, in a three-year study period, Giménez-Bañón et al. [42] observed a significant 31% reduction in total phenols in grapes treated with nano-MeJ during the 2020 season. The same authors, when Monastrell grapes were treated with urea and nano-urea, observed yearly fluctuations in total phenol levels within grape cell walls, influenced by different treatments. These decreases may indicate a cell wall weakening, as TPcw is involved in the structural cross-linking.

With regard to protein content, the highest values were found in MeJ-treated grapes (Table 4) and the lowest in MeJ-ChNPs. ChNPs showed intermediate values. These proteins are likely to be structural components that reinforce the wall during berry expansion (extensin), since they are considered to form a fibrillar network [36]. Most cell wall proteins are cross-linked within the polysaccharide network, which makes them difficult to extract [43]. Different authors have reported varying results depending on the treatment, variety, and vintage. Paladines-Quezada et al. [2] found that the highest protein levels in 2015 were observed in Monastrell grapes treated with benzothiadiazole (BTH) and in Cabernet Sauvignon grapes treated with methyl jasmonate (MeJ). In 2016, however, increased protein levels were only detected in Monastrell grapes treated with MeJ, while the Merlot variety consistently showed the lowest protein concentrations across both years when treated with either elicitor. The same authors observed that the MeJ treatment applied during veraison was the only one that increased the protein concentration in Monastrell grapes. Finally, Apolinar-Valiente et al. [44], in a study conducted on various hybrids derived from Monastrell, observed a lower protein content. Therefore, it is reasonable to suggest that the reduced protein levels in the grape skins of most progeny genotypes—compared to the Monastrell parent—may indicate weaker structural rigidity, potentially facilitating the degradation of the cell wall material in these grapes.

### 2.7. Microscopic (Transmission Optical Microscopy) Characterisation of Berry Tissue Structure

Figure 2 illustrates the microscopic comparison between control and treated grape skins. Optical microscopy revealed the structural arrangement of the exocarp and mesocarp, emphasising differences in the number of cellular layers from the cuticle to the pulp, depending on the treatment. The outermost layers contained densely packed cells with thick walls, whereas the inner layers were composed of larger cells with a more relaxed arrangement. Optical microscopy enabled the evaluation of the morphology of the cells in the most external layers of grape skins (Figure 2).

Microscopic analysis revealed that grape skins treated with MeJ-ChNPs and subsequently with ChNPs developed more cell layers and thicker cell walls compared to untreated grapes, with this effect being less pronounced in grapes treated solely with MeJ. Notably, cellular integrity was maintained throughout harvest, suggesting that these treatments may reinforce the structural durability of epidermal and hypodermal tissues during ripening. These observations align with the data shown in Table 3, which indicate a higher percentage of cell wall material in grapes treated with ChNPs and MeJ-ChNPs.

Evident differences in the number of cell layers from the cuticle to the pulp were observed between control and treated samples. Such morphological changes in skin cell wall composition may significantly impact the winemaking process. At harvest, grapes treated with MeJ-ChNPs and ChNPs exhibited more extensive layering and thicker cell walls than those treated with MeJ alone or the control group, which showed thinner walls and a sharper transition between skin and pulp tissues.

Physical characteristics of grape skins—such as berry skin hardness, skin thickness, number of cell layers, and cell wall thickness—are closely linked to anthocyanin extractability and depend heavily on cell wall composition [45]. Therefore, differences in cell wall morphology and structure may explain the variability in anthocyanin extraction efficiency during winemaking [28].

### 2.8. Chromatic Parameters in Wines

The release of grape phenolic compounds corresponds to the diffusion of specific cell wall components into the fermentation as a consequence of grape pomace degradation and maceration occurring during alcoholic fermentation, and extractability is modulated by several factors, including the variety and geographic origin of the grape [14], the agronomic practices and ripening stage [33], as well as the structural and compositional characteristics of the berry skin cell walls [46].

Grape skin cell walls are a protective barrier for the extraction of phenolic compounds that consist mainly of polysaccharides, lipids, proteins, and phenolics bound to the cell wall components [36]. When cell walls are disrupted, polyphenol extraction from the vacuole occurs rapidly, with two simultaneous phenomena taking place: (1) binding to the cell walls through hydrogen bonding and hydrophobic interactions, and (2) chemical modifications. Both processes can contribute to the observed losses of anthocyanins and tannins, reflected in the differences between the quantities measured in grape berries and those found in wines, lees, and pomace [47]. Therefore, differences in cell wall composition could impact the extractability of phenolic compounds and relate to the differences observed in skin degradation during fermentation [46], playing a crucial role in the vinification process.

The different chromatic parameters measured in wines from control and treated grapes are shown in Table 5. With respect to TPwines, these were higher in wines from the control grapes compared to those from treated grapes. These results may be linked to the findings in Section 2 regarding extractability parameters, where it was already observed that control grapes exhibited higher levels of extractable polyphenols, and Section 3, where it was revealed that the highest concentration of TPcw was found in the cell walls of the control grapes. Additionally, analytical evidence has demonstrated that treated grapes possess thicker cell walls, which could hinder the extraction process—particularly for tannins and other phenolic compounds—compared to control wines. Other authors, such as Garrido-Bañuelos et al. [46], have also attributed a greater extractability of phenolic compounds to the degree of grape ripeness. Riper grapes tend to facilitate phenolic extraction due to increased de-esterification in the pectin layer, which likely enhances the access to the inner layers of the pomace and promotes the subsequent release of these compounds. In our study, we also observed that at harvest, control grapes exhibited higher °Brix values compared to treated grapes, indicating a more advanced stage of ripeness.

With regard to CI, we did not obtain significant differences between wines from control and treated grapes; however, the wines from the treatment with MeJ-ChNPs showed the lowest CIElab parameter values, indicating more intense wine properties and bluer and redder colours. In contrast to our findings, Garrido-Bañuelos et al. [46], in a study carried out with grapes at two maturity stages observed that wines made from 21°Brix grapes showed significantly lower colour and phenolic parameters than the dry wines from 23°Brix and 25°Brix grapes. The opposite was observed in our study where the grapes treated with MeJ-ChNPs had a lower soluble solid content than those harvested with other treatments (or no treatment).

Additionally, wines from MeJ-ChNPs showed the highest concentration of total anthocyanins. These results are in agreement with those reported by Paladines-Quezada et al. [2], who observed that in 2016, the total anthocyanin content increased significantly in Monastrell and Merlot wines from MeJ- and BTH-treated grapes, but not in Cabernet Sauvignon wines. This could be explained by the fact that these grapes showed the highest ApH1 values; therefore, it would be necessary that, during the maceration process, a chemical reaction occurred that completely destroyed the cell wall where these compounds were included. Authors such as Amrani-Joutei et al. [41] stated that the release of anthocyanins depends on the integrity of the vacuole membrane, the tonoplast, and the cell walls, which play a key role, as they appear to limit the extraction of polyphenols by binding them. On the other hand, the presence or absence of extraction of anthocyanins during the winemaking process may be marked by the anthocyanin profile of the studied variety, as coumaroylated anthocyanins have a higher affinity for grape cell walls than non-acylated anthocyanins [47], and cyanidin-3-glucoside, a non-acylated anthocyanin, has a higher affinity with less methylated pectins [48]. Finally, during winemaking, anthocyanins may be degraded into lower-molecular-weight compounds, converted to pyranoanthocyanins, or involved in reactions with tannins, leading to the formation of adducts, such as pigmented tannin-anthocyanin and tannin-ethyl-anthocyanin, or colourless anthocyanin-tannin, which could also have an influence on the detection of a lower amount of total anthocyanins in the produced wines.

Concerning total tannins, no statistically significant differences were observed in tannin concentrations between control wines and those produced from treated grapes. However, the highest tannin levels were found in the control samples. This finding aligns with the results of Garrido-Bañuelos et al. [46], who reported that wines made from grapes at 21°Brix retained more non-extracted tannins in the fermented pomace than those made from grapes at 23° or 25°Brix.

Tannins are located either freely within vacuoles or bound to cell wall components such as proteins and polysaccharides, with up to 70% bound to polysaccharides [41,47]. During alcoholic fermentation, polysaccharides extracted from grapes and yeasts can modulate tannin aggregation, thereby influencing the colloidal stability of the wine. Previous studies have shown that tannin concentrations in wine are often lower than expected [49]. This discrepancy may be due not only to the limited extraction of tannins from grape skins and seeds, but also to the adsorption of tannins onto suspended skin and pulp cell walls in the must, which subsequently precipitate during settling [29]. Interactions between proanthocyanidins and cell wall polysaccharides have been extensively documented [50], and their oenological implications have been explored in depth [51,52]; these may also affect their extractability during the alcoholic fermentation process.

Additional factors influencing the extractability of phenolic compounds during alcoholic fermentation include temperature and increasing ethanol levels, which facilitate the extraction of anthocyanins and tannins from grape skins. This process also depends on the chemical structure of the cell walls. Anthocyanins appear to enhance the extraction of tannins, although the underlying mechanism remains unclear. Losses exceeding 50% have been reported for both anthocyanins and tannins after their extraction from skins in model [47]; however, the relative contributions of irreversible binding versus other losses caused by chemical modifications are still poorly understood. Previous studies have shown that certain grape varieties, despite having high anthocyanin content in their skins, exhibit limited extractability—such is the case with the Monastrell variety [33]. Indeed, the concentration of anthocyanins and the colour intensity in the resulting wines did not correlate with the total anthocyanin levels found in the grapes, as a significant portion remained in the skins even after a 15-day maceration period [28].

Therefore, although the morphology and composition of the cell wall are determining factors in the extractability of phenolic compounds during the winemaking process, many other factors must be taken into account, as the extraction process is complex and has not been fully clarified yet.

### 2.9. Correlation Analysis Between Cell Wall Components and Chromatic Parameters

A Pearson correlation analysis (Table 6) was conducted for the control and each treatment to identify potential relationships between the various cell wall components and the extraction of phenolic compounds as influenced by the treatments. The Pearson correlation coefficient assesses the strength and direction of the linear relationship between two variables, ranging from −1 (perfect negative correlation) to +1 (perfect positive correlation).

The correlation analysis between grape skin cell wall composition and wine chromatic parameters reveals complex and treatment-dependent interactions. Although strong correlations were observed, only some reached statistical significance, and the direction of these relationships varied according to the elicitor applied. In the control group, hemicellulose and TPcw showed high positive correlations with anthocyanins, while uronic acids were positively associated with tannins. These findings suggest that, under natural conditions, polysaccharide-rich matrices facilitate pigment release, whereas uronic acids contribute to tannin retention. However, Giménez-Bañón et al. [42] reported a negative correlation between cellulose and total phenolics and a positive correlation between hemicellulose and colour intensity in Monastrell grapes treated with MeJ and nano-MeJ, highlighting that cell wall–phenolic interactions are highly context-dependent.

When MeJ was applied alone, correlations shifted toward negative values. Uronic acids correlated negatively with total phenolics and tannins, and cellulose showed a strong inverse relationship with anthocyanins. These results align with those of Paladines-Quezada et al. [2], who found a high negative correlation between uronic acids and total phenolics in grapes treated with MeJ and BTH. Conversely, Giménez-Bañón et al. [42] observed a positive correlation between uronic acids and total phenolics, underscoring the variability of responses across studies. Taken together, our results indicate that MeJ treatment reinforces the cell wall, thereby hindering the extraction of phenolic compounds. Consequently, higher proportions of uronic acids in the cell wall are associated with reduced phenolic extractability.

Nanoparticle treatments produced a different outcome. ChNPs showed a strong positive correlation between uronic acids and anthocyanins, suggesting that this elicitor promotes pigment biosynthesis and enhances extractability. In contrast, the combined MeJ-ChNPs treatment exhibited positive correlations between TPcw and both anthocyanins and tannins, but negative correlations between uronic acids and anthocyanins and tannins, as well as between cellulose and total phenolics and tannins. These results suggest that while the combination stimulates pigment biosynthesis, the reinforced polysaccharide matrix limits diffusion into the must. Similar trends were reported by Giménez-Bañón et al. [17], who observed strong negative correlations between uronic acids and wine tannins under urea and nanotreatments, indicating that increased uronic acid content reduces phenolic extractability during winemaking.

Overall, these correlations demonstrate that the extractability of phenolic compounds can be partly explained by the composition of the grape skin cell wall. More importantly, they confirm that elicitor treatments actively remodel cell wall morphology, thereby altering the release of phenolics and pigments that define wine chromatic properties. Chitosan nanoparticles alone appear to favour anthocyanin accumulation and extraction, while MeJ reinforces the cell wall and suppresses pigment release. The combined MeJ-ChNPs treatment generates a paradoxical effect: enhanced biosynthesis of anthocyanins coupled with reduced extractability, leading to wines with distinctive but complex colour profiles.

### 2.10. Multivariable Analysis: Principal Components

Principal component analysis (PCA) was performed to gain insight into how the various treatments influenced the cell wall composition and the extraction of phenolic compounds in the wine (Figure 3). It was observed that the variance contribution rates of PC1 and PC2 were 41.71% and 25.97%, respectively. The first two PCs showed a cumulative variance contribution rate of 66.68% (>65%); thus, they were considered the most informative from the entire sample.

Analysis of the PCA biplot revealed distinct patterns in the distribution of treatments along the first two principal components. PC1, which corresponds to the horizontal axis and accounts for a substantial proportion of the total variance, delineated a clear separation between control and treated samples. Control samples were predominantly located on the positive side of PC1, influenced by high loadings of TPwine, TPpH3.6, ApH3.6, and CI. In contrast, the MeJ and ChNPs treatments were situated on the negative side of this component, primarily associated with elevated Cellulose content. This axis effectively captures a fundamental divergence in cell wall composition and phenolic characteristics between untreated and treated wine samples.

On the other hand, PC2, representing the vertical axis, introduced an additional dimension of differentiation. The MeJ-ChNPs treatment was uniquely positioned on the positive side of PC2, driven by increased levels of UAs and ApH1. Meanwhile, the remaining treatments (MeJ, ChNPs) and the control group cluster together on the negative side, characterised by higher Protein content. This component underscored the distinctive biochemical profile of the combined MeJ-ChNPs treatment, suggesting a synergistic effect not observed in the individual treatments.

Overall, the control samples are positioned on the right side of the graph, showing a clear contrast with the treated samples on the left. Interestingly, the MeJ and ChNPs treatments are closely similar to each other but stand out from the MeJ-ChNPs treatment, which emerges as the most distinct one.

## 3. Materials and Methods

### 3.1. Chemicals

Chitosan (medium molecular weight, degree of deacetylation ≥ 75%), methyl jasmonate, Tween 80, Bradford reagent, gallic acid, 3,5-dimethylphenol, pure galacturonic acid, phenol, and Folin–Ciocalteu reagent were purchased from Sigma-Aldrich (St. Louis, MO, USA). Pure acetone, 96% ethanol, 1 N sodium hydroxide, glacial acetic acid (99%), 0.1 N hydrochloric acid, tartaric acid, and 98% sulfuric acid were obtained from Panreac (Barcelona, Spain). The enzymatic glucose assay kit was supplied by TDI (Tecnología Difusión Ibérica S.L., Gavà, Spain). Bovine serum albumin (BSA), fraction V, used for protein quantification, was acquired from Roche Diagnostics GmbH (Mannheim, Germany). Ultrapure water was produced using a Milli-Q purification system (Millipore Corp., Bedford, MA, USA).

### 3.2. Preparation and Characterisation of Methyl Jasmonate-Loaded Chitosan Nanoparticles

A 5-fold stock suspension of methyl jasmonate (MeJ)-loaded chitosan nanoparticles (MeJ-ChNPs), equivalent to 10 mM MeJ, was prepared through a two-step process, as illustrated in Figure 1A.

The procedure involved (1) emulsification of chitosan with MeJ and Tween 80, followed by (2) ionic gelation using tripolyphosphate (TPP), based on the method described by Shetta et al. [16], with some modifications. Briefly, 1% *w/v* chitosan solutions were prepared by dissolving chitosan in 0.75% *v/v* acetic acid containing 1% *w/v* Tween 80, under magnetic stirring at room temperature for 24 h. Subsequently, undissolved chitosan was removed by centrifugation at 10,000× *g* for 30 min using an Eppendorf Centrifuge 5810R (Eppendorf AG, Hamburg, Germany). After that, the clear supernatant phase was transferred to a glass beaker. At a later stage, MeJ was added and emulsified with an IKA T25 Ultraturrax (IKA, Dusseldorf, Germany) at a speed of 13,000 rpm for 3 min until a homogeneous milky emulsion was formed. Subsequently, to induce ionic gelation of chitosan, TPP solution was slowly dropped into the emulsion while the suspension was kept under Ultraturrax mixing for 10 min (Ch/TPP ratio: 5:1 *w*/*w*). Afterward, the suspension was maintained under magnetic stirring for 2 h to allow the release of entrapped air bubbles from the nanoparticle suspension. It was then stored at 4 °C until use. Unloaded nanoparticles, used as a control, were prepared following the same protocol but without the addition of MeJ. The hydrodynamic properties of the nanoparticles—including particle size, polydispersity index, and zeta potential—were evaluated using a Zsizer Advance Ultra Red Label (Malvern Panalytical, Malvern, UK). Particle morphology was examined by scanning electron microscopy (Thermo Fisher Scientific Inc., Waltham, MA, USA), following the procedures and using the instruments previously described by Pérez-Lloret et al. [53].

### 3.3. Experimental Field

The study was conducted during the 2024 growing season in an experimental vineyard located in Cehegín (Murcia, Spain) (latitude: 38.11179; longitude: −1.6808). The trials were performed on Vitis vinifera L. cv. Monastrell grafted onto Richter 110 rootstock, trained in a bilateral cordon system with a planting density of 3 × 0.8 m (between and within rows, respectively). Phenology dates were the following: bud-break in April; flowering in June; veraison in August and finally harvest in September. Four treatments were applied at veraison, one week later, and again two weeks after the initial application, using 10 vines per replicate. The treatments included: (i) control (water); (ii) 2 mM MeJ; (iii) ChNPs (chitosan nanoparticles); and (iv) MeJ-ChNPs (equivalent to 2 mM MeJ). All solutions contained Tween 80 (0.1% *v*/*v*), and each vine was sprayed with 200 mL of the corresponding treatment.

Regarding meteorological information, total precipitation, average temperature, maximum temperature and minimum temperature during the ripening period (July–September) were 12.5 mm, 21.4 °C, 27.1 °C and 19.8 °C, respectively, in 2024 (data were recorded by the Agricultural Information System of the Region of Murcia (SIAM)).

### 3.4. Physico-Chemical Parameters of Grapes

A sampling of approximately 300 g was taken at harvest to carry out the different physico-chemical analyses. The must was obtained by crushing the samples using a Robot coupé model Gt 550 (Montcea Les Mines, Bourgone, France). The sample was then centrifuged for 15 min at 4400 rpm. Physico-chemical parameters were evaluated, including total soluble solids (°Brix), total acidity (TA), pH, and the concentrations of tartaric and malic acids. Total soluble solids were measured using an Abbé-type refractometer (Atago RX-5000, Tokyo, Japan), while pH and total acidity were determined with an automatic titrator (Metrohm, Herisau, Switzerland) using 0.1 N NaOH. Tartaric and malic acid concentrations were quantified using a MIURA ONE automatic analyser (TDI Tecnología Difusión Ibérica, SL, Barcelona, Spain). Additionally, the weight of 100 berries was recorded as an indicator of size grape grain.

### 3.5. Extractability Parameters

The phenolic potential of the grapes was determined using the method outlined by Saint Cricq et al. [54], which involves a 4 h maceration at two pH levels (3.6 and 1.0). Anthocyanin content in both extracts—representing extractable and total anthocyanins (ApH3.6 and ApH1, respectively)—was quantified by measuring absorbance at 520 nm at the respective pH values. Extractable phenolic content (TPpH3.6) was assessed by measuring absorbance at 280 nm in the solution adjusted to pH 3.6. The original solution at pH 3.2 was replaced with one at pH 3.6, which better reflects the characteristics of must from the Jumilla area. The extractability indexes were then calculated by using Equations (1) and (2), respectively:



(1)
$$Index\;of\;Cellular\;Maturity\;\left(\%\right)\left(IMC\right)=\left[\frac{ApH1-ApH3.6}{TPpH3.6}\right]\times 100$$


(2)
$$Seed\;Maturity\;Index\;\left(\%\right)\left(SMI\right)=\left(TPpH3.6-\frac{\frac{ApH3.6x40}{1000}}{TPpH3.6}\right)\times 100$$



### 3.6. Isolation of Cell Wall Material (CWM)

Cell walls were isolated using the alcohol-insoluble solids separation method, as described by De Vries et al. [55], with some modifications that were already proposed in our previous report [42]. Briefly, the grapes (100 g for each treatment) were frozen at −20 °C. The skins were then peeled with a scalpel and stored at −20 °C. Subsequently, they were lyophilised using a Cryodos 50 (IMA-TELSTAR, Terrassa, Spain) and then pulverised in a ball mill (Vibratory Ball Mill Pulverisette 0, Cryo-box, FRITSCH, Idar-Oberstein, Germany). Each pulverised grape skin sample was boiled in 50 mL of water for 5 min to inactivate the enzymes and then centrifuged. Afterwards, the supernatant was removed. Each of the samples was then washed several times (30 min at 40 °C with 70% ethanol, centrifuged, and separated from the supernatant) until the soluble sugars were removed. The alcohol-insoluble solids, named as ICW (isolated cell wall material), were washed twice with 96% ethanol and once with acetone. Finally, the samples were dried overnight at 20 °C.

The percentages of dry skin (%Skin) and the cell wall (%CWM) in the grapes were calculated by using Equations (3) and (4), respectively:
(3)$$\%\;Skin=\frac{weight\;of\;lyophilised\;grape\;skins}{weight\;of\;fresh\;grapes\;}\times 100$$
(4)$$\%\;CWM=\frac{weight\;of\;ICW\;}{weight\;of\;lyophilised\;grape\;skins\;}\times 100$$

### 3.7. Analysis of the Composition of the Grape Skin Cell Wall

The following variables were measured on isolated CWM (10 mg for each assay) and in quadruplicate for each of the treatments: proteins, phenolic compounds (TPcw), cellulosic glucose (cellulose), non-cellulosic glucose (hemicellulose), and uronic acids (UAs). The determination of each of the components was performed using the methodologies described below.

#### 3.7.1. Proteins and Phenolic Compounds

A sample of CWM (10 mg) was treated with 1 M NaOH at 100 °C for 10 min and then centrifuged in an Eppendorf 5810 R (Hamburgo, Germany) for 5 min at 10,000 rpm. Both analyses were performed on the supernatant using a UV/visible spectrophotometer, Model 1600-UV. Proteins were analysed using the colourimetric Coomassie Brilliant Blue assay [56], with the bovine serum albumin (BSA) fraction V used to create the calibration curve. Results were expressed as mg BSA per gram of cell wall.

The determination of phenolic compounds (TPcw) was carried out using the colourimetric Folin–Ciocalteu reagent test. The calibration curve was generated using gallic acid, and the results were expressed as milligrams of gallic acid per gram of cell wall.

#### 3.7.2. Uronic Acids, Cellulose, and Hemicellulose

Uronic acids (UAs) and total glucose were analysed after pre-hydrolysis at 30 °C for 1 h with 72% aqueous sulfuric acid, followed by hydrolysis with 1 M sulfuric acid at 100 °C for 3 h. UAs, as an indicator of pectins, were measured in the supernatant using a colourimetric 3,5-dimethylphenol assay [57] with a UV/visible spectrophotometer, model 1600-UV (Shimadzu Corporation, Kyoto, Japan). D-(+)-galacturonic acid monohydrate was used as the standard for the calibration curve.

Cellulose and hemicellulose were determined in the same supernatant using a segmented continuous-flow analyser (QuAAtro 39, SEAL analytical, Norderstedt, Germany). The analyser measures reducing sugars based on their reaction with the copper neocuproine complex chelate in a basic medium at 90 °C, yielding a yellow complex chelate whose absorbance is measured at 460 nm. Calibration was performed using standards ranging from 0.1 to 1 g/L of glucose.

### 3.8. Optical Microscopy

Mature berry samples were processed for microscopic analysis following the protocol outlined by Amrani Joutei et al. [41], with minor modifications. Freshly harvested berries were sectioned into small fragments of tissue (1 mm^2^, approximately) and fixed in McDowell reagent (25% *v/v* glutaraldehyde and 40% *v/v* formaldehyde in 0.2 M cacodylate buffer) at 4 °C for 24 h. Post-fixation was performed using osmium tetroxide under identical temperature conditions for an additional period of 2.5 h. Samples were thoroughly rinsed with phosphate buffer between fixation steps. Dehydration was achieved through a graded ethanol series (30% to 95% *v*/*v*), followed by three washes in absolute ethanol containing copper sulphate, after which they were embedded in SPURR-type resin. Semi-thin sections were stained with toluidine blue and examined under an optical microscope.

### 3.9. Vinification

Grapes were hand-harvested into boxes and transported to the experimental winery located in Jumilla, Murcia (Spain). Vinification was carried out following a traditional protocol using 50 L stainless steel tanks. During destemming and crushing, potassium metabisulphite was added at a rate of 50 mg/kg. Commercial yeast (Saccharomyces cerevisiae, SafCEno™ NDA 21; 20 g/100 kg) was then inoculated. The must’s acidity was adjusted with tartaric acid to reach 5.5 g/L. Alcoholic fermentation was conducted at 25 °C, with temperature and density monitored twice daily, along with aerated pump-overs. Maceration lasted for 14 days. Once alcoholic fermentation was complete (wine density ≤ 0.995 kg/L), the wine was drained, and the grape skins were pressed using a pneumatic press. The resulting wines were racked and cold-stabilised.

### 3.10. Spectrophotometric Variables in Wines

Spectrophotometric parameters were measured in wine samples at the end of alcoholic fermentation, with each measurement performed in triplicate. Total anthocyanin content (TAwine) was determined using a colorimetric method based on the Puissant–Léon technique employing an automatic MIURA ONE analyser (TDI Tecnología Difusión Ibérica, SL, Barcelona, Spain). Colour intensity (CI) and total polyphenol content (TPwine) were assessed using a Shimadzu UV/Visible spectrophotometer model 1600PC (Shimadzu Corporation, Kyoto, Japan). CI was calculated as the sum of absorbance values at 620 nm (blue), 520 nm (red), and 420 nm (yellow) in undiluted wine samples [58] while TP_wine was determined by measuring absorbance at 280 nm

### 3.11. Statistical Analysis

Significant differences among treatments for each variable were evaluated using analysis of variance (ANOVA). Duncan’s multiple range test was applied to compare the means, with differences considered statistically significant at *p* < 0.05. Additionally, Pearson correlation coefficients were calculated between cell wall component parameters and the chromatic attributes of the wines. Finally, principal component analysis (PCA) was performed on the various cell wall components, extractability and chromatic parameters using the statistical software Statgraphics Centurion 18, Version 18.1.14 (B 2023 Statgraphics Technologies, Inc., The Plains, VA, USA)

## 4. Conclusions

This study provides new insights into the effects of elicitor treatments—methyl jasmonate (MeJ), chitosan nanoparticles (ChNPs), and their combination (MeJ-ChNPs)—on the Monastrell grape variety, highlighting their capacity to modulate grape maturation, cell wall morphology and phenolic extractability. Whereas control grapes contained higher phenolic concentrations at a pH of 3.6, MeJ-ChNPs-treated grapes demonstrated greater extractability at a pH of 1, indicating an enhanced phenolic accumulation due to the elicitor. The treatments, particularly those involving nanoparticles, delayed ripening and induced significant cell wall thickening, as evidenced by increased CMI and CWM values confirmed through optical microscopy. Regarding cell wall composition, biochemical analyses revealed that MeJ-ChNPs treatment enhanced uronic acid and cellulose content, whereas control grapes exhibited higher levels of hemicellulose, total polyphenols (TPcw), and proteins, a trend partially mirrored in the MeJ treatment.

These structural modifications had direct consequences for wine composition and chromatic properties. Although wines derived from control grapes contained higher total polyphenols (TPwine), MeJ-ChNPs wines displayed reduced CIELAB colour parameters but a marked increase in anthocyanin concentration (TAwine). This suggests that elicitation, particularly with MeJ-ChNPs, promotes anthocyanin biosynthesis and enhances pigment accumulation, potentially improving wine quality despite lower overall polyphenolic content. On the other hand, correlation analyses further demonstrated that modifications in cell wall composition directly influenced phenolic diffusion during fermentation, while PCA revealed distinct clustering of treatment groups, with MeJ-ChNPs emerging as the most differentiated and unique profile.

Taken together, these findings underscore the potential of nanoparticle-based elicitors—especially MeJ-ChNPs—as innovative tools to improve wine quality by modulating grape physiology and phenolic composition. However, the accompanying morphological and biochemical changes in the grape cell wall must be carefully considered from a technological perspective, as they may influence extraction dynamics and winemaking processes. Future studies should explore the scalability of these treatments in vineyard conditions and assess their long-term impact on wine stability, sensory attributes, and consumer acceptance.

## Figures and Tables

**Figure 1 plants-14-03817-f001:**
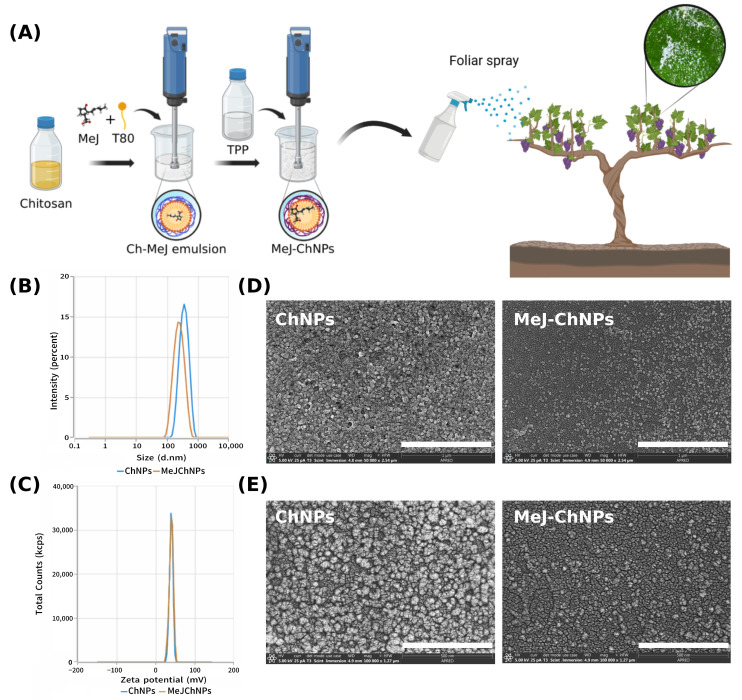
(**A**) Scheme of the preparation of methyl jasmonate-loaded chitosan nanoparticles and the field treatment. (**B**) Hydrodynamic size distributions of the ChNPs and MeJ-ChNPs. (**C**) Zeta potential distribution of the ChNPs and MeJ-ChNPs. (**D**,**E**) Scanning electron micrographs of the ChNPs and MeJ-ChNPs at different magnifications (Scale bar = 1 µm (**D**) or 500 nm (**E**)).

**Figure 2 plants-14-03817-f002:**
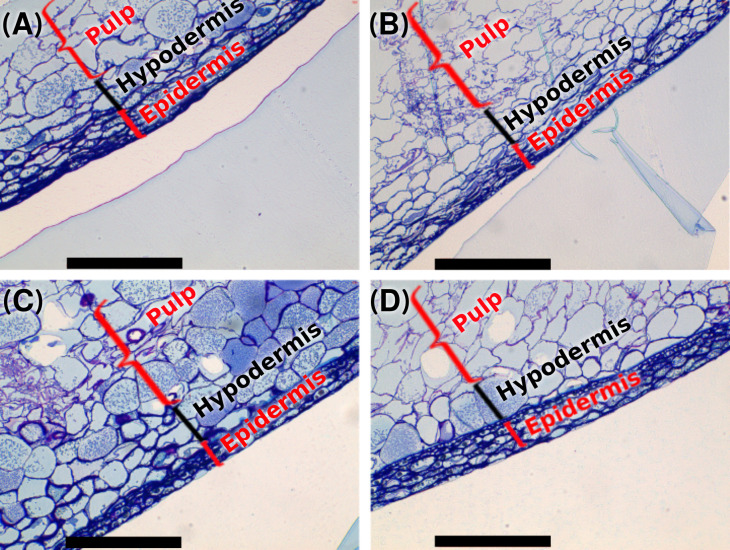
Optical microscopy images of sections of the skins of the treated and control grapes. (**A**) Control grapes (**B**) MeJ-treated grapes (**C**) ChNPs treated grapes and (**D**) MeJ-ChNPs treated grapes.

**Figure 3 plants-14-03817-f003:**
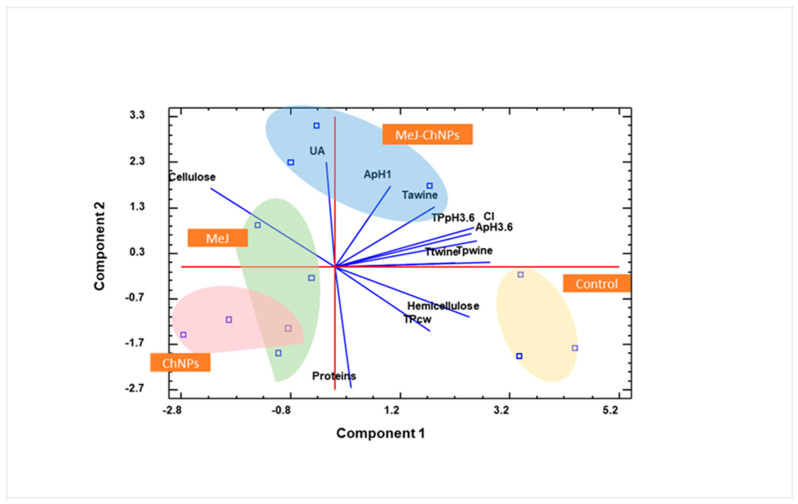
PCA biplot of different components related to extractability, cell wall composition and chromatic parameters in wines. Abbreviations: MeJ: Methyl jasmonate; ChNPs: Chitosan nanoparticles; MeJ-ChNPs: Chitosan nanoparticles loaded with MeJ; TPcw: Total phenol in cell wall; TPwine: Total phenols in wines; CI: Colour intensity; TAwine: Total anthocyanins in wines; TTwines: Total tannins in wines; ApH3.6: Anthocyanin extracted at pH 3.6; TPpH3.6: Total polyphenols extracted at pH 3.6; ApH1: Anthocyanin extracted at pH 1; UA: Uronic acids.

**Table 1 plants-14-03817-t001:** Physico-chemical parameters in control and treated grapes at harvest time.

	Control	MeJ ^b^	ChNPs	MeJ-ChNPs
Soluble solids (°Brix)	24.61 ± 0.56 b ^a^	23.60 ± 0.56 a	23.94 ± 0.74 a	23.58 ± 0.10 a
Total acidity ^c^	2.58 ± 0.06 a	2.74 ± 0.07 b	3.90 ± 0.25 c	3.96 ± 0.16 c
pH	4.01 ± 0.03 c	3.98 ± 0.02 bc	3.95 ± 0.09 ab	3.92 ± 0.04 a
Tartaric Acid (g/L)	5.07 ± 0.09 a	5.63 ± 0.10 b	5.49 ± 0.15 b	5.64 ± 0.08 b
Malic acid (g/L)	2.20 ± 0.04 a	2.37 ± 0.08 b	2.18 ± 0.10 a	2.19 ± 0.04 a
Berry weight ^d^	1.99 ± 0.03 b	1.91 ± 0.07 ab	1.82 ± 0.11 a	1.91 ± 0.13 ab

^a^ Different letters in the same row indicate significant differences according to Duncan test (*p* < 0.05). ^b^ Abbreviations: MeJ: Methyl jasmonate; ChNPs: Chitosan nanoparticles; MeJ-ChNPs: Chitosan nanoparticles loaded with MeJ. ^c^ expressed as g/L tartaric acid. ^d^ expressed as g.

**Table 2 plants-14-03817-t002:** Extractability parameters in control and treated grapes at harvest moment.

	**Control**	**MeJ ^b^**	**ChNPs**	**MeJ-ChNPs**
ApH3.6 ^b^ (mg/L)	423.77 ± 4.36 c^a^	385.48 ± 9.43 b	328.96 ± 4.42 a	377.06 ± 4.62 b
TPpH3.6	52.13 ± 0.35 c	43.63 ± 0.80 ab	40.05 ± 0.68 a	45.23 ± 0.30 b
ApH1 (mg/L)	627.83 ± 13.37 ab	611.30 ± 14.22 ab	551.66 ± 15.86 a	688.20 ± 8.29 b
CMI (%)	32.17 ± 1.69 a	38.41 ± 0.82 ab	39.37 ± 3.74 b	44.94 ± 2.43 b
SMI (%)	67.45 ± 0.87 a	64.68 ± 0.84 a	67.10 ± 0.26 a	66.57 ± 1.23 a

^a^ Different letters in the same row indicate significant differences according to Duncan test (*p* < 0.05). ^b^ Abbreviations: MeJ: Methyl jasmonate; ChNPs: Chitosan nanoparticles; MeJ-ChNPs: Chitosan nanoparticles loaded with MeJ. ApH3.6: Anthocyanin extracted at pH 3.6; TPpH3.6: Total polyphenols extracted at pH 3.6; ApH1: Anthocyanin extracted at pH 1; CMI: Cellular maturity index; SMI: Seed maturity index.

**Table 3 plants-14-03817-t003:** Skin and cell wall isolated from control and treated grapes.

	**% Skin**	**ICW mg/g Skin**	**% Cell Wall**	**% Skin/% Cell Wall**
Control	7.63 ± 0.02 b ^a^	226.00 ± 1.75 a	22.60 ± 0.11 a	0.34 ± 0.01 a
MeJ ^b^	6.56 ± 0.03 a	237.00 ± 3.56 ab	23.74 ± 0.09 ab	0.28 ± 0.02 ab
ChNPs	7.17 ± 0.01 b	280.00 ± 2.55 b	27.96 ± 0.09 b	0.25 ± 0.01 b
MeJ-ChNPs	6.46 ± 0.02 a	306.00 ± 1.27 c	30.64 ± 0.23 c	0.21 ± 0.02 b

^a^ Different letters in the same column indicate significant differences according to Duncan test (*p* < 0.05). ^b^ Abbreviations: MeJ: Methyl jasmonate; ChNPs: Chitosan nanoparticles; MeJ-ChNPs: Chitosan nanoparticles loaded with MeJ.

**Table 4 plants-14-03817-t004:** The grape skin cell wall composition in control and treated grapes.

	**Control**	**MeJ^b^**	**ChNPs**	**MeJ-ChNPs**
**UA s**	194.09 ± 6.91 a^a^	193.56 ± 2.78 a	188.56 ± 2.34 a	211.07 ± 1.36 b
**Hemicellulose**	217.12 ± 5.42 b	193.94 ± 2.35 a	192.34 ± 2.09 a	192.96 ± 1.16 a
**Cellulose**	197.88 ± 5.95 a	215.09 ± 7.31 b	209.53 ± 1.22 ab	218.13 ± 4.44 b
**TPcw**	108.09 ± 11.41 b	49.54 ± 4.97 a	49.52 ± 4.66 a	52.39 ± 3.32 a
**Proteins**	58.56 ± 2.33 b	54.68 ± 2.25 b	52.22 ± 2.48 ab	43.61 ± 5.21 a

^a^ Different letters in the same column, indicate significant differences according to Duncan test (*p* < 0.05). ^b^ Abbreviations: MeJ: Methyl jasmonate; ChNPs: Chitosan nanoparticles; MeJ-ChNPs: Chitosan nanoparticles loaded with MeJ; UAs: uronic acids (expressed as mg of galacturonic acid g^−1^ of cell wall); hemicellulose and cellulose (expressed as mg g^−1^ of cell wall); Total phenols (expressed as mg of gallic acid g^−1^ of cell wall), proteins (expressed as mg of BSA equivalents g^−1^ of cell wall).

**Table 5 plants-14-03817-t005:** Chromatic parameters in wines at the end of alcoholic fermentation.

	**Control**	**MeJ ^b^**	**ChNPs**	**MeJ-ChNPs**
TPwine	41.44 ± 3.65 b ^a^	35.22 ± 1.70 a	35.42 ± 2.40 a	37.21 ± 2.78 ab
CI	8.92 ± 0.65 a	8.14 ± 0.56 a	8.17 ± 0.61 a	9.00 ± 0.55 a
L*	19.95 ± 1.68 ab	21.85 ± 1.38 ab	22.29 ± 1.97 b	19.05 ± 1.07 a
a*	52.48 ± 1.80 ab	54.42 ± 1.34 b	54.86 ± 1.88 b	51.41 ± 1.00 a
b*	32.79 ± 1.91 ab	34.71 ± 1.19 b	35.55 ± 1.81 b	31.48 ± 1.34 a
C*	61.89 ± 2.54 ab	64.55 ± 1.77 b	65.37 ± 2.56 b	60.28 ± 1.56 a
H*	31.98 ± 0.61 ab	32.52 ± 0.26 bc	32.93 ± 0.43 c	31.47 ± 0.57 a
TTwine	1245.39 ± 77.83 a	1058.29 ± 76.98 a	1123.77 ± 99.00 a	1166.80 ± 86.40 a
TAwine	373.33 ± 14.43 bc	351.33 ± 24.19 b	307.00 ± 24.58 a	388.67 ± 10.02 c

^a^ Different letters in the same row indicate significant differences according to Duncan test (*p* < 0.05). ^b^ Abbreviations: MeJ: Methyl jasmonate; ChNPs: Chitosan nanoparticles; MeJ-ChNPs: Chitosan nanoparticles loaded with MeJ. TPwine: Total phenols in wine; CI: Colour intensity; L*: Lightness; a*: red–green axis; b*: yellow–blue axis; C*: Chroma; H*: Hue angle; TTwine: Total tannins in wine; TAwine: Total anthocyanins in wine.

**Table 6 plants-14-03817-t006:** Pearson correlation between cell wall components and chromatic parameters.

**Treatment**	**Cell Wall Components**	**Wine Chromatic Parameters**			
	** **	**TP_wine_**	**CI**	**Anthocyanins**	**Tannins**
Control	UA s	0.8848	0.5412	−0.5	0.9993 *
	Glucose	0.4297	0.8183	0.8856	−0.0045
	Hemicellulose	−0.2049	0.3031	0.986	−0.6091
	Cellulose	0.6172	0.1504	−0.8102	0.8974
	Proteins	−0.0214	−0.51	0.8709	0.2015
	TP_Cw_	−0.2987	0.2094	0.9653	−0.683
MeJ ^b^	UAs	−0.9619	−0.8939	−0.5122	−0.9776
	Glucose	−0.7574	−0.9995 *	−0.8261	−0.9754
	Hemicellulose	−0.867	−0.3025	0.258	−0.5311
	Cellulose	−0.0287	−0.6966	−0.9732	−0.4952
	Proteins	−0.6448	0.0411	0.5723	−0.2099
	TP_Cw_	−8416	−0.2555	0.3049	−0.489
ChNPs	UAs	0.1378	0.4589	0.9614 *	−0.703
	Glucose	−0.3235	−0.0566	0.672	−0.5512
	Hemicellulose	−0.1833	−0.3272	0.5378	−0.1165
	Cellulose	−0.3078	0.0701	0.5735	−0.6067
	Proteins	−0.1925	0.2597	−0.0336	−0.0731
	TP_Cw_	0.7984	0.7239	0.0665	0.7971
**MeJ-ChNPs**	UAs	−0.8333	−0.454	−0.9607	−0.9354
	Glucose	−0.9973 *	−0.8327	−0.7011	−0.9886
	Hemicellulose	−0.0067	−0.4972	0.7581	0.2159
	Cellulose	−0.9512	−0.6767	−0.8508	−0.9970 *
	Proteins	−0.0214	−0.51	8709	0.2015
	TP_Cw_	0.836	0.4584	0.9593	0.9371

* Statistical significant differences according to Duncan test (*p* < 0.05). ^b^ Abbreviations: MeJ: Methyl jasmonate; ChNPs: Chitosan nanoparticles; MeJ-ChNPs: Chitosan nanoparticles loaded with MeJ; UAs: Uronic acids; TPcw: Total phenol in cell wall; TPwine: Total phenols in wines; CI: Colour intensity; TAwine: Total anthocyanins in wines; TTwines: Total tannins in wines.

## Data Availability

Data is contained within the article.

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
