# Peer review of "Linking Grape Cell Wall Composition and Phenolic Release to Wine Quality: Effects of Methyl Jasmonate-Loaded Chitosan Nanoparticles in Monastrell"

_plants, 2025, doi:10.3390/plants14243817_

Round 1

Reviewer 1 Report

Comments and Suggestions for Authors

This study examined the impact of three elicitor treatments—conventional methyl jasmonate, chitosan nanoparticles, and a combination of both—on the Monastrell grape variety. The topic is very current, mainly due to climate change and difficulties in phenolic maturation of grapes in different regions of the world.
The work has a broad bibliography that supports the results obtained.
The methods used were adequate and are able to characterize both the grapes and the wines.
The absence of data regarding individual phenolic composition, obtained by chromatographic techniques, represents an important limitation of the study, as such information would be fundamental to identifying which groups of compounds were effectively impacted by the application of the elicitors. This is suggested for future work and can be added to the conclusions.
Check in the text and change - the unit °C is separated from the number by a space according to international metrology standards.
A negative aspect of the work is the performance of the experiment in only one harvest, which may compromise the reliability of the results, due to climatic variability between years; Genotype × environment interaction and scientific reproducibility.
With this in mind, I suggest that the authors add a paragraph in the materials and methods section with the climate data for the growing season (2024) for the study region, which would help to complement and explain the results obtained.

Author Response

Review Report Form #1:

Comments and Suggestions for Authors

This study examined the impact of three elicitor treatments—conventional methyl jasmonate, chitosan nanoparticles, and a combination of both—on the Monastrell grape variety. The topic is very current, mainly due to climate change and difficulties in phenolic maturation of grapes in different regions of the world. The work has a broad bibliography that supports the results obtained. The methods used were adequate and are able to characterize both the grapes and the wines.

  1. The absence of data regarding individual phenolic composition, obtained by chromatographic techniques, represents an important limitation of the study, as such information would be fundamental to identifying which groups of compounds were effectively impacted by the application of the elicitors. This is suggested for future work and can be added to the conclusions.

Thanks for the suggestion to the reviewers. Indeed, we have carried out this study and it will be the subject of another article that we will publish soon.

  1. Check in the text and change - the unit °C is separated from the number by a space according to international metrology standards.

Thank you for the observation, which has already been corrected in the manuscript.

  1. A negative aspect of the work is the performance of the experiment in only one harvest, which may compromise the reliability of the results, due to climatic variability between years; Genotype × environment interaction and scientific reproducibility.

Thanks for the observation, it is true that the study would be more complete with at least two vintages, but because we have had to modify the formulation of the nanoparticles, we believe from our previous experience that the results found are relevant enough to be published.

With this in mind, I suggest that the authors add a paragraph in the materials and methods section with the climate data for the growing season (2024) for the study region, which would help to complement and explain the results obtained.

It has been included in the paper the following paroagraph: Regarding meteorological information, total precipitation, average temperature, maximum temperature and minimum temperature during the ripening period (July–September) were 12.5 mm, 21.4 °C, 27.1°C and 19.8°C respectively in 2024 (data were recorded by the Agri-cultural Information System of the Region of Murcia (SIAM)).

Reviewer 2 Report

Comments and Suggestions for Authors

This study makes substantial contributions to viticulture and enology by addressing critical challenges in grape quality improvement and sustainable agriculture. Its core significance lies in the innovative integration of nanotechnology with phytohormone application, offering a promising solution to the long-standing limitations of methyl jasmonate (MeJ) use. By encapsulating MeJ in chitosan nanoparticles (MeJ-ChNPs), the research effectively mitigates MeJ’s volatility and low solubility, enhancing its stability and elicitation efficiency. The combination of research materials with nanotechnology demonstrates certain innovation, and the workload is considerable. The manuscript is generally well-structured, but the following issues need to be addressed:

  1. The title requires revision. It should more closely reflect the study’s focus on the link between grape cell wall modulation, phenolic release, and wine quality, while enhancing academic appeal and precision.
  2. The abstract lacks a clear statement of the research objectives.
  3. The current keywords are not sufficiently precise and comprehensive.
  4. The introduction fails to adequately justify the selection of MeJ as the treatment agent and lacks basic information about the grape material. It should clarify why MeJ is suitable for grape quality improvement and specify that Monastrell is a wine grape variety, laying a foundation for the study’s relevance to enology.
  5. The results and analysis section lacks logical hierarchy and appears disjointed. Since wine-related data are included, the title should explicitly reflect the connection to wine quality. It is recommended to restructure the results by grouping related findings to enhance readability. Additionally, there are figure number inconsistencies (e.g., Line 373 refers to Figure 2 or Figure 1), which require systematic verification and correction.
  6. The "Materials and Methods" section is structured illogically. Following the thinking of agricultural science researchers, plant materials (Monastrell grapes, vineyard conditions) should be described first, followed by treatment materials (preparation of MeJ-ChNPs, treatment protocols) to ensure a logical research workflow.

Recommended for publication following necessary revisions.

Author Response

Review Report Form #2:

Comments and Suggestions for Authors

This study makes substantial contributions to viticulture and enology by addressing critical challenges in grape quality improvement and sustainable agriculture. Its core significance lies in the innovative integration of nanotechnology with phytohormone application, offering a promising solution to the long-standing limitations of methyl jasmonate (MeJ) use. By encapsulating MeJ in chitosan nanoparticles (MeJ-ChNPs), the research effectively mitigates MeJ’s volatility and low solubility, enhancing its stability and elicitation efficiency. The combination of research materials with nanotechnology demonstrates certain innovation, and the workload is considerable. The manuscript is generally well-structured, but the following issues need to be addressed:

  1. The title requires revision. It should more closely reflect the study’s focus on the link between grape cell wall modulation, phenolic release, and wine quality, while enhancing academic appeal and precision.

As the reviewer suggests, the title has been changed as follows in order to link cell wall modulation, phenolic release and wine quality: Linking grape cell wall composition and phenolic release to wine quality: Effects of methyl jasmonate-loaded vhitosan nanoparticles in Monastrell

  1. The abstract lacks a clear statement of the research objectives.

As suggested by the reviewer, the abstract of the article has been rewritten, clarifying the objectives of the work as well as the results obtained in it.

  1. The current keywords are not sufficiently precise and comprehensive.

As other reviewer has suggested two new key words has been added: cell wall composition; chitosan nanoparticles.

  1. The introduction fails to adequately justify the selection of MeJ as the treatment agent and lacks basic information about the grape material. It should clarify why MeJ is suitable for grape quality improvement and specify that Monastrell is a wine grape variety, laying a foundation for the study’s relevance to enology.

The introduction has already pointed out why it would increase the quality of the grapes, as explained in the following paragraph:

“Amongst chemical elicitors, MeJ is a natural phytohormone that has been reported to up-regulate endogenous levels of health-promoting compounds by increasing phenolics, an-thocyanins, and flavonoid biosynthesis, in addition to increasing antioxidant levels [1].”

Furthermore, there are numerous studies in which MeJ is used as an elicitor not only in vineyards but also in other types of crops as an elicitor to increase the synthesis of secondary metabolites:

Gil-Muñoz, R., Giménez-Bañón, M.J., Bleda-Sánchez, J.A., Moreno-Olivares, J.D. The Impact of Two Elicitors and Harvest Ripening Stage on the Quality of Monastrell Grapes and Wines. Biomolecules, 15(4), 474 (2025). https://doi.org/10.3390/biom15040474.

Garde-Cerdán, T., González-Lázaro, M., Sáenz de Urturi, I., Marín-San Román, S., Baroja, E., Rubio-Bretón, P., Pérez-Álvarez, E.P. Application of Methyl Jasmonate and Methyl Jasmonate + Urea in Tempranillo Vines: Influence on Grape Phenolic Compounds. American Journal of Enology and Viticulture, 73(2), 2022. https://doi.org/10.5344/ajev.2022.22026.

García-Pastor, M.E., Serrano, M., Guillén, F., Castillo, S., Martínez-Romero, D., Valero, D., Zapata, P.J. Methyl jasmonate effects on table grape ripening, vine yield, berry quality and bioactive compounds depend on applied concentration. Scientia Horticulturae, 247, 380–389 (2019). https://doi.org/10.1016/j.scienta.2018.12.012

  1. The results and analysis section lacks logical hierarchy and appears disjointed. Since wine-related data are included, the title should explicitly reflect the connection to wine quality. It is recommended to restructure the results by grouping related findings to enhance readability. Additionally, there are figure number inconsistencies (e.g., Line 373 refers to Figure 2 or Figure 1), which require systematic verification and correction.

As the reviewer suggested, the title of the work has been modified, which we believe now gives coherence and meaning to the results and discussion section of the manuscript, making the reason for combining cell wall results with results obtained in wines clearer.

Regarding Line 373, as show up in the manuscript, refers to Figure 2:

“Figure 2. Optical microscopy images of sections of the skins of the treated and control grapes. A) 373 Control grapes B) MeJ-treated grapes C) ChNPs treated grapes and D) MeJ-ChNPs treates grapes”

  1. The "Materials and Methods" section is structured illogically. Following the thinking of agricultural science researchers, plant materials (Monastrell grapes, vineyard conditions) should be described first, followed by treatment materials (preparation of MeJ-ChNPs, treatment protocols) to ensure a logical research workflow.

Although the author appreciate the reviewer's suggestion, the authors do not believe there is any inconsistency in the presentation of the materials and methods, since we first listed the reagents used in all the analyses and then the nanoparticle synthesis, as we have done in other published works, before proceeding to describe the experimental setup and the analyses performed.

Examples of other published papers:

Giménez-Bañón MJ, Paladines-Quezada DF, Moreno-Olivares JD, Bleda-Sánchez JA, Fernández-Fernández JI, Parra-Torrejón B, Ramírez-Rodríguez GB, Delgado-López JM, Gil-Muñoz R. Methyl Jasmonate and Nanoparticles Doped with Methyl Jasmonate affect the Cell Wall Composition of Monastrell Grape Skins. Molecules. 2023 Feb 3;28(3):1478. doi: 10.3390/molecules28031478. PMID: 36771144; PMCID: PMC9921610.

Giménez-Bañón, M.J.; Paladines-Quezada, D.F.; Moreno-Olivares, J.D.; Bleda-Sánchez, J.A.; Fernández-Fernández, J.I.; Parra-Torrejón, B.; Ramírez-Rodríguez, G.B.; Delgado-López, J.M.; Gil-Muñoz, R. Methyl Jasmonate and Nanoparticles Doped with Methyl Jasmonate affect the Cell Wall Composition of Monastrell Grape Skins. Molecules 2023, 28, 1478. https://doi.org/10.3390/molecules28031478

Reviewer 3 Report

Comments and Suggestions for Authors

Dear authors,

     I hope you are doing well!

     The aim of this research was to evaluate the effects of MeJ treatments applied conventionally and in chitosan nanoparticles loaded with MeJ on the structural components of the cell walls of Monastrell grape skins, together with their impact on the extraction of phenolic compounds in the resulting wines. It was innovated and necessary to be researched, and the experiment design was reasonable.

    However some suggestions must be given because there were some issues needing debating.

    1. In section abstract, some key results weren't introduced clearly such as the vagueness in the so-called changes or modifications and affections and so on.

   2. Regarding to the keywords, they need regulating, and suggest to delete the word morphology, then add the words such as "cell wall composition" and "chitsan nanoplasticles".

       3. Regarding to the title 2, it had better be modified into "Results and Discussions".

     4.  Please check the full text carefully because there were some unreasonable expressions such as "Quality parameters in grapes" and non-italic Latin names and so on. "Quality parameters in grapes" must be changed into "Quality parameters in grape fruits".

      5.  That change the words "at the moment of harvest" into the words "at harvest" would be more reasonable.

       6.  Regarding to all tables, the letters symbolizing the significance of the differences among the different treatments must be sequenced in turns of the size of the values, yet the sequence of the letters was to the contrary at present.  Additionally please add the "±SD" after the means. 

     7. In section 2.9, the linear correlations weren't be analysed clearly and deeply, only compared with the former reports.

         8. In section 2.10, it is necessary to check the results because that the first two PCs showed a cumulative variance contribution rate of 91.78% seemed false. It is more important for the principle components analysis results to be analysed deeply, at least the Directionalities of the two PCs need pointing out.

        9. In section materials and methods, please add the phenological periods of the grape, the replication times of the treatments, the sampling method and the samples dealing methods.

         10. Section conclusions needs further sublimation. 

Comments on the Quality of English Language

      For examples,   "Quality parameters in grapes" must be changed into "Quality parameters in grape fruits", and that change the words "at the moment of harvest" into the words "at harvest" would be more reasonable.

      Non-italic Latin names need modification.

Author Response

Review Report Form #3:

The aim of this research was to evaluate the effects of MeJ treatments applied conventionally and in chitosan nanoparticles loaded with MeJ on the structural components of the cell walls of Monastrell grape skins, together with their impact on the extraction of phenolic compounds in the resulting wines. It was innovated and necessary to be researched, and the experiment design was reasonable.

  1. In section abstract, some key results weren't introduced clearly such as the vagueness in the so-called changes or modifications and affections and so on.

As suggested by the reviewer, the abstract of the article has been rewritten, clarifying the objectives of the work as well as the results obtained in it.

  1. Regarding to the keywords, they need regulating, and suggest to delete the word morphology, then add the words such as "cell wall composition" and "chitsan nanoplasticles".

As the reviewer suggests, the word “morphology” has been deleted and the words “cell wall composition” and “chitosan nanoparticles” have been added.

  1. Regarding to the title 2, it had better be modified into "Results and Discussions".

As the reviewer suggests, the title 2 has been modified in the manuscript.

  1. Please check the full text carefully because there were some unreasonable expressions such as "Quality parameters in grapes" and non-italic Latin names and so on. "Quality parameters in grapes" must be changed into "Quality parameters in grape fruits".

As the reviewer suggested, the full text has been revised and the corresponding changes have made.

  1. That change the words "at the moment of harvest" into the words "at harvest" would be more reasonable.

As the reviewer suggests, the change has been done in the manuscript.

  1. Regarding to all tables, the letters symbolizing the significance of the differences among the different treatments must be sequenced in turns of the size of the values, yet the sequence of the letters was to the contrary at present. Additionally please add the "±SD" after the means.

As we have published in numerous journals (https://doi.org/10.3390/biom15040474;doi: 10.3390/molecules26061689; https://doi.org/10.1016/j.foodchem.2018.08.009, and so on), the sequencing of the letters in this sense to specify the significant differences when applying an ANOVA.

With respect to ±SD, it has been added in all tables.

  1. In section 2.9, the linear correlations weren't be analysed clearly and deeply, only compared with the former reports.

As suggested by the reviewer, the abstract of the article has been rewritten, clarifying the objectives of the work as well as the results obtained in it.

  1. In section 2.10, it is necessary to check the results because that the first two PCs showed a cumulative variance contribution rate of 91.78% seemed false. It is more important for the principle components analysis results to be analysed deeply, at least the Directionalities of the two PCs need pointing out.

As suggested by the reviewer, the contribution of the two principal components, PC1 and PC2, to the variance was verified and corrected, as there was an error; it was not 91.78% but 66.68%. Furthermore, the paragraph referring to the contribution of each principal component to the final observed results was rewritten.

  1. In section materials and methods, please add the phenological periods of the grape, the replication times of the treatments, the sampling method and the samples dealing methods.

Phenological periods are added in the text: ). “Phenology dates were the following: bud-break in April; flowering in June; veraison in August and finally harvest in September”

Replication times of the treatment are explained in the text: “were applied at veraison, one week later, and again two weeks after”

Sampling dealing has been added in the text between lines 623-626

  1. Section conclusions needs further sublimation.

The conclusions of the work have been rewritten, going into more detail about the results shown throughout the manuscript.

  1. Comments on the Quality of English Language. For examples, "Quality parameters in grapes" must be changed into "Quality parameters in grape fruits", and that change the words "at the moment of harvest" into the words "at harvest" would be more reasonable.

As the reviewer suggests, the changes have been done in the manuscript.

  1. Non-italic Latin names need modification.

These modifications have been done in the manuscript.

Reviewer 4 Report

Comments and Suggestions for Authors

I have attached the file with my comments.

Author Response

Review Report Form #4

The title suggests a broader aspect of the grape physiology on one side, and the other side is more specific. The article is about the grape berry skin cell wall composition, not in general grape cell wall composition. On the other hand, it is not so important to write in the title that it is in Monastrell grape variety, because it has limited international importance.

Detailed comments

  1. The section of the Abstract is well written. I suggest modifying part of the last sentence (row 32) as this study did not result in the elicitor effect as general changes in grapevine physiology, but moderated the grapeskin morphology and components.

As the reviewer suggests, the last part of the abstract has been modified.

  1. The section Introduction contains related citations. I only suggest removing the rice root-related citation (rows 79-81) and replacing it with the grape skin-related ones.

I have not found any reference that implies the cultivation of the vine between the high nitrogen levels have been shown to suppress the expression of genes involved in lig-nin and cellulose biosynthesis: Therefore, the authors have decided to leave this quote in the manuscript.

  1. All the chemicals used in this study are sufficient for the planned examinations. The preparation of MeJ-ChNPs is clear. I suggest extending the Experimental field description with the vineyard plant protection protocol and a short description of the climatic conditions of the year. Both of the mentioned information are important, even to the authors, as they mentioned in the introduction that the external environment heavily influences the cell wall morphology and components.

Regarding climatic conditions, a short paragraph has been added in the manuscript.

  1. In row 605 should be more precise that the measured parameters of the grape berries. Also, for this section important to note that the 100 berries' weight does not reflect the yield load of the vine stock.

Yes, the reviewer is actually right, berry weight tells you berry size but not production, so we have eliminated it from the manuscript.

  1. It would be worth explaining what the phenolic potential (row 615) is. I'd better like to measure the exact amount of the phenolic compound.

This methodology precisely measures the phenolic extractability potential that the grape would have, in two different pH conditions, and that is what we want to measure to then calculate the two ripening indices that will explain the ease of extraction of the anthocyanins and tannins from the seed. As for the exact amount of phenolic compounds, we have analyzed the different families by liquid chromatography, but that will be the subject of another article.

6.I guess cell wall thickness was examined under optical microscopy; however, how it was measured should be added to that section (rows 674-685).

Although the reviewer suggested specifying the use of the optical microscope, which consists of the following:

“When a sample is observed under a light microscope, it is first prepared on a glass slide, often covered with a coverslip. Illumination is provided by a light source that passes through a condenser, focusing the beam onto the specimen. The objective lenses magnify the image, while the eyepiece provides the final enlargement. Finally, focus is adjusted using coarse and fine knobs, allowing clear visualization of cell wall.”

The authors considered that such a level of detail is necessary in a scientific article.

  1. In the section of Results, I suggest removing the preparation of MeJ-ChNPs and adding it to the Materials and Methods section.

Thank you for your suggestion, but the authors consider the Preparation and Characterization of the nanoparticles to be an important part of the work, given that the nanoformulations have an intrinsic interest and the whole process has been successfully scaled up.

  1. My question is about section 2.2. How was the harvest time decided? And it is only one factor that influences the ripening stage of the berries among a lot of others, besides the treatments.

The harvest time was decided when the optimal technological ripening was reached by control grapes, which in the case of Monastrell is usually around 14ºBé or 25ºBrix.

  1. Regarding Table 1, how is it possible that the total acid content is less than the tartaric acid and malic acid content together? Please check your measured data! Also in Table 1, berry size (width and length in mm) is not equal to berry weight (g)! Please correct it!

Total acidity in must analyzes is usually lower than the sum of tartaric and malic acids because total acidity is not measured as the simple sum of grams of each acid, but as tartaric acid equivalents expressed in terms of their neutralization capacity. This means that it depends on the stoichiometry, the number of available protons and the pKa of each acid, so the values ​​are not directly additive.

In other scientific papers, we have put the berry size like this, but perhaps berry weight is more correct, so as the reviewer suggests we have changed it in the manuscript.

  1. Regarding Table 3, the %skin, if I understood well, is a % of the total berry weight, and the % cell wall is a % of the berry skin. If I am right, please write clearly on it.

As shown in equations (3) and (4), the formulas clearly specify how both parameters are calculated:

  1. Regarding Table 5, the abbreviations need to have the L*, a*,b*,c*, H* meanings.

The following abbreviations have been added in Table 5:

L*: Lightness; a*: Red–Green Axis; b*:Yellow–Blue Axis; C*:Chroma (color intensity or saturation); H*: Hue angle (tone).

  1. In the section Conclusion, I do not understand in row 722 the “pH 1”, as none of the samples had that low pH, or I misunderstood something? Otherwise, most of the concluded results are correctly written down.

To clarify this part, pH 1 was used in the methodology to determine the extractability of phenolic compounds (Section 3.4).

Round 2

Reviewer 3 Report

Comments and Suggestions for Authors

Dear authors,

      I hope you are doing well!

      I feel honoured that my suggestions could help you to improve the academic quality of the manuscript.

      However it's still wrong that you answered to the issue 9th. I didn't mean how many times the experiment treatments were be treated, but mean how many times the same name plots in the field experiment design were, so please continue to modify. 

Author Response

The experiments were carried out for one year, since due to the results obtained (in other parameters such as individual phenolic compounds, aromatic compounds, amino acids...) that were not as expected, it was decided to change the formulation, and this year we have made applications with the same components but a different mode of nanoparticle synthesis.